# Donor Genetic Predisposition to High Interleukin-10 Production Appears Protective against Acute Graft-Versus-Host Disease

**DOI:** 10.3390/ijms232415888

**Published:** 2022-12-14

**Authors:** Gaurav Tripathi, Rutvij A. Khanolkar, Rehan M. Faridi, Amit Kalra, Poonam Dharmani-Khan, Meer-Taher Shabani-Rad, Noureddine Berka, Andrew Daly, Jan Storek, Faisal M. Khan

**Affiliations:** 1Cumming School of Medicine, University of Calgary, Calgary, AB T2N 1N4, Canada; 2Department of Pathology and Laboratory Medicine, Calgary, AB T2L 1N4, Canada; 3Alberta Precision Laboratories, Calgary, AB T2L 2K8, Canada; 4Alberta Health Services, Calgary, AB T2N 4L7, Canada

**Keywords:** cytokine gene polymorphism, IL-10, graft-versus-host disease (GVHD), hematopoietic cell transplantation (HCT), biomarkers

## Abstract

The persistence of graft-versus-host disease (GVHD) as the principal complication of allogeneic hematopoietic cell transplantation (HCT) demonstrates that HLA matching alone is insufficient to prevent alloreactivity. We performed molecular and functional characterization of 22 candidate cytokine genes for their potential to improve matching in 315 myeloablative, 10/10 HLA-matched donor–recipient pairs. Recipients of a graft carrying the -1082GG IL10 gene promoter region variant had a three-fold lower incidence of grade II–IV acute GVHD compared to IL10-1082AA graft recipients (SHR = 0.25, *p* = 0.005). This was most evident in matched unrelated donor (MUD) transplants, where the greatest alloreactivity is expected. IL10-1082GG transplants did not experience an increased incidence of relapse, and, consequently, overall survival was two-fold higher in IL10-1082GG MUD transplants (HR = 0.17, *p* = 0.023). Longitudinal post-transplant measurements demonstrated that -1082GG is a high-IL10-producing and -expressing genotype with attenuated CD8^+^ T-cell reconstitution. High post-transplant donor chimerism in T- and myeloid-cells (>95%) confirmed a predominant donor, rather than recipient, genotype effect on immune function and aGVHD. To date, this is the first study to report corroborating genome-to-cellular evidence for a non-HLA donor immunogenetic variant that appears to be protective against GVHD. The incorporation of IL10 variants in donor selection criteria and clinical-management decisions has the potential to improve patient outcomes.

## 1. Introduction

Despite improvements in human leukocyte antigen (HLA) matching, graft-versus-host disease (GVHD) remains the most substantial cause of morbidity and mortality following allogeneic hematopoietic cell transplantation (HCT). As outcomes often vary significantly between patients with similar prognostic factors and transplant-related characteristics, both stochastic and non-HLA genetic factors may contribute to inter-individual differences. There has been growing interest in the discovery of novel genetic factors that may be used to further refine allogeneic donor selection criteria with killer-immunoglobulin-like receptors (KIR), minor histocompatibility antigens, and cytokine-related genes being among the most studied candidates [1,2,3].

Cytokine-related variants that impact mRNA expression and serum protein levels appear to be particularly promising given that the dysregulation of cytokine networks plays a key role in GVHD pathogenesis [4,5]. Interleukin-10 (IL-10) is primarily a regulatory cytokine which plays a key role in the suppression of proinflammatory cytokines and Th1 lymphocytes, while stimulating the proliferation and differentiation of regulatory T-cells (Tregs) and Th2 cells [6]. IL-10 is also a known suppressor of CD8^+^ T-cell activity [7], with IL-10 overproduction having been shown to suppress acute GVHD pathogenesis [8,9]. Although numerous studies have reported on the association between IL-10 gene variants and transplant outcomes [1,2,10,11,12,13,14,15,16,17,18,19,20,21], most studies have reported on clinical associations with limited or no corroborating analysis of serum cytokine levels, mRNA expression, or impact on immune reconstitution. Additionally, most well-powered studies to date have been conducted in the setting of non-myeloablative HCT, and found an association between recipient IL-10 genotype and aGVHD [11,12,13,14,15,16,20,21,22,23,24,25,26,27,28].

The current study evaluated the impact of donor IL-10 genotype in the setting of myeloablative HCT in a large cohort of 10/10 HLA-matched donor–recipient pairs. Recipients of a graft with a single nucleotide polymorphism (SNP) in the donor IL-10 gene promoter region were found to be protected against the development of grade II–IV acute GVHD. Furthermore, as no study, to date, has longitudinally assessed the mechanism of this association, we evaluated mRNA, protein, and immune reconstitution of T-cell subsets to generate the first comprehensive dataset that elucidates a mechanistic picture for the role of donor IL-10 genotype in GVHD pathogenesis. Unlike previous studies, which primarily reported the association in the recipient genotype, the finding that the donor IL-10 genotype may protect against the development of aGVHD introduces the possibility of improving outcomes through the incorporation of the IL-10 genotype into donor–recipient matching.

## 2. Results

### 2.1. IL-10 Promoter Region Variant Protects against aGVHD without Increasing Relapse

A total of 22 donor SNPs across 13 cytokine and cytokine receptor genes were evaluated in a discovery cohort of 171 donors for their association with clinically significant GVHD (sGVHD; defined as grade II–IV aGVHD or moderate–severe cGVHD). In multivariate modeling, sGVHD was significantly associated only with the A/G variant at the -1082 position of the IL-10 promoter region (rs1800896). This association was subsequently confirmed in a validation cohort consisting of 144 donors (Appendix A). Among total study patients (*n* = 315), recipients of a graft with a GG polymorphism in IL-10-1082 position had a >three-fold lower incidence of grade II–IV aGVHD (9% vs. 30%, *p* = 0.005, SHR = 0.25) and >two-fold lower incidence of sGVHD (25% vs. 58%, *p* = 0.001, SHR = 0.33) compared to recipients of an IL-10-1082AA genotype graft (Figure 1C). Although there was a trend towards an association for moderate–severe cGVHD, this did not reach statistical significance (Figure 1).

In general, the incidence of GVHD is expected to be greater in patients with a matched unrelated donor (MUD) than in those with a matched related donor (MRD) due alloreactivity linked with other immunogenetic factors such as mismatches at minor-histocompatibility loci [29,30]. Therefore, we theorized that the protective effect of the IL-10-1082GG allele would be most pronounced in the MUD setting. In transplants performed with an MUD graft (*n* = 157), recipients of an IL-10-1082GG genotype graft appeared to have significantly better protection against the development of grade II–IV aGVHD (*p* = 0.008, SHR = 0.15) and sGVHD (*p* = 0.021, SHR = 0.27) (Figure 1). However, in patients with an MRD graft (*n* = 158), there was no statistically significant difference in GVHD incidence between recipients of the -1082GG vs. AA genotype (Figure 1).

Additionally, although it was expected that a decrease in the GVHD effect would be associated with a concordant reduction in the graft-versus-leukemia (GVL) effect, no difference was observed in the incidence of relapse between recipients of an IL-10-1082GG vs. AA graft in the MUD, MRD, or total patient groups (Figure 2). 

### 2.2. Improved Overall Survival in IL-10-1082GG MUD Graft Recipients

As aGVHD and relapse are the primary causes of mortality after HCT, we sought to determine if protection from aGVHD without an increase in relapse corresponded to improved survival outcomes. Among MUD transplants, IL-10-1082GG graft recipients had a >two-fold greater overall survival compared to IL-10-1082AA graft recipients (87% vs. 43%; *p* = 0.023, SHR = 5.92) (Figure 2). This association was not statistically significant in the MRD recipients or total study patients (Figure 2). 

### 2.3. No Impact of Recipient IL-10 SNPs on GVHD

To date, investigations of IL-10 variants have largely found an association between recipient, but not donor, SNPs with GVHD [11,12,13,14,15,16,20,21,22,23,24,25,26,27,28]. As these studies were primarily conducted in the non-myeloablative MRD transplant setting, we sought to assess whether this association was also present in the myeloablative MUD setting using pre-transplant DNA samples from all 315 recipients. No difference was observed in the incidence of aGVHD or cGVHD based on IL-10 SNP, regardless of donor type (Appendix A). 

### 2.4. IL-10-1082GG Is a High-IL10-Expressing and -Producing Genotype

Although multiple groups have previously reported on the association between IL-10 SNPs and aGVHD, remarkably few studies have correlated clinical findings with IL-10 proteins levels [20], and, to our knowledge, no comprehensive and longitudinal assessment of both mRNA and protein levels has been conducted to date. To support our clinical findings with corroborating transcriptomic and proteomic data, we measured IL-10 *mRNA* and serum protein levels longitudinally post-transplant. 

Serum IL-10 levels were found to be elevated in IL-10-1082GG graft recipients at 1-month post-transplant (*p* = 0.006), but not at other measured timepoints (Figure 3A). Analysis of 1-month mRNA expression showed that *IL-10* transcripts were four-fold elevated in IL-10-1082GG recipients at the 1-month timepoint (*p* = 0.011) (Figure 3B). 

When comparing recipients who did vs. did not later develop grade II–IV aGVHD, IL-10 levels were significantly elevated in those that did not develop aGVHD at 1 week, 1 month, and 2 months post-transplant (*p* < 0.01, *p* < 0.001, *p* = 0.04) (Figure 3D). Additionally, IL-10 levels were significantly elevated at 1 week and 1 month post-transplant (*p* = 0.034, *p* = 0.012) in recipients who did vs. did not later develop sGVHD (Figure 3D). There was no significant difference in IL-10 levels at any timepoint between individuals who did vs. did not develop moderate–severe cGVHD (Figure 3D). These results are consistent with previously presented findings showing that the IL-10-1082GG genotype is protective against grade II–IV aGVHD and sGVHD, but not against moderate–severe cGVHD (Figure 1).

### 2.5. High Donor Chimerism Supports a Predominant Donor-Genotype Effect on IL-10 Levels and Protection against aGVHD

In the setting of myeloablative HCT, it is expected that the majority of hematopoietic cells in the recipient post-transplant are donor-derived. However, complete immune ablation may not always be achieved, and in these cases, the role of the recipient’s immune system cannot be automatically discounted. In response to inflammation, IL-10 production is chiefly mediated by monocytes and lymphocytes [31]. Therefore, we performed chimerism assays on sorted T- and myeloid-cells at 1 month post-transplant. High (>95%) chimerism was present in both T-cells and myeloid cells at 1 month post-transplant (Figure 3C). This supports our findings that it was the donor, rather than the recipient, IL-10 genotype which was responsible for the observed differences in post-transplant IL-10 levels and the incidence of aGVHD.

### 2.6. More Dominant T-Lymphocyte Phenotype in Recipients of an IL-10-1082AA Graft

Next, we aimed to determine how early reconstitution of T-cell subsets differs based on IL-10 genotype. Our group has previously shown that aGVHD is preceded by high counts of CD4^+^ and CD8^+^ T-cells [32]. We have also shown that, in renal transplantation, low-IL10-producing genotype grafts had greater T-cell infiltration and inflammation [33]. Therefore, we evaluated the impact of IL-10-1082 A/G on the relative abundance of total CD3^+^, CD4^+^, and CD8^+^ T-cells at 1 week, 1 month, 2 months, and 3 months post-transplant.

Recipients of a low-IL10-producing graft (IL-10-1082AA) were found to have a significantly elevated proportions of CD8^+^ T-cells at 1 week, 1 month, and 2 months post-transplant (*p* = 0.012, *p* = 0.035, *p* = 0.029) compared to recipients of a high-IL10-producing graft (IL-10-1082GG) (Figure 3E). There was no significant difference in total CD3^+^ or CD4^+^ T-cells counts at any of the measured timepoints (Figure 3E). 

## 3. Discussion

GVHD is the most substantial iatrogenic complication of HCT. As the immunologic reaction preceding GVHD begins to occur only hours to days after transplant [34,35], and because GVHD is difficult to treat once it has clinically manifested, preventing alloreactivity through improved immunogenetic matching is highly desirable. In this study, recipients of a graft with an IL-10 promoter -1082GG polymorphism were found to have a significantly lower incidence of grade II–IV aGVHD. Importantly, with the apparent protection against aGVHD, there was no concomitant increase in the incidence of relapse. Further analysis showed that the reduced incidence of aGVHD was most pronounced in recipients of MUD grafts carrying the IL-10-1082GG genotype, translating into an improved overall survival. This study also represents the first to include a comprehensive longitudinal assessment of IL-10 mRNA, serum protein, and T-cell subsets. We found that IL-10-1082GG is a high-IL10-producing and -expressing genotype, with attenuated reconstitution of CD8^+^ T-cells, which are known to play a key role in aGVHD pathogenesis [32]. This difference was most pronounced early after transplant (1 week, 1 month, and 2 months) but not at later timepoints. This is consistent with the finding that IL-10-1082GG appears to be protective against aGVHD (median onset of 1.5 months post-transplant), but not cGVHD (median onset of 4.6 months post-transplant). Perhaps most importantly, our study demonstrates the importance of the donor, rather than recipient, IL-10 genotype in protection from aGVHD. Taken together, these findings represent the first comprehensive dataset providing evidence for a non-HLA immunogenetic variant that may be used to improve donor–recipient matching.

IL-10 is primarily a regulatory cytokine that plays a key role in the suppression of pro-inflammatory Th1 lymphocytes and the production of IFN-γ, IL-6, IL-1, IL-12, and TNF-α [6]. IL-10 also stimulates the proliferation and differentiation of regulatory T-cells (Tregs) and Th2 cells [6]. Although no absolute distinction exists between T-cell subsets that mediate aGVHD vs. cGVHD, it is generally recognized that aGVHD is a T-dominant reaction while B-cells are more important in the pathogenesis of cGVHD [36]. IL-10 is a known suppressor of CD8^+^ T-cell activity [7], with IL-10 overproduction having been shown to suppress aGVHD pathogenesis [8,9]. In a previous study by our group on renal transplantation, we have shown that recipients of high-IL10-producing genotype grafts had decreased infiltration of T-cells and macrophages, with no impact on B-cells [33]. This corresponds with current study in that a genetic predisposition towards high IL-10 production is associated with protection against aGVHD, but not cGVHD. 

Like all cytokines, the production of IL-10 is primarily under genetic control [37,38], which makes it plausible that promoter-region variants can alter production. However, unlike many other cytokines, IL-10 production is not constitutive and, rather, is induced in response to inflammation. This explains why studies have found no statistically significant difference in the serum IL-10 levels between healthy volunteers with a GG vs. AA genotype [39]. Therefore, the immune-dampening effects of IL-10 are expected to be most prominent in the setting of active GVHD and only minimally present in its absence. This explains why, in the current study, recipients of a high-IL10-producing graft had a reduced incidence of aGVHD, but no concomitant increase in the incidence of relapse (i.e., dampened graft-versus-leukemia effect). 

An important strength of this study compared to previous reports is the use of a homogeneously treated cohort that received uniform transplant conditioning, GVHD prophylaxis, and supportive care. We also employed the use of an independent discovery and validation cohort, which reduces the risk of finding false-positive associations. Additionally, all patients received 10/10 HLA-matched grafts. The purpose of this restriction was to isolate and identify the effect of non-HLA immunogenetic factors in GVHD pathogenesis. However, even in the absence of HLA-mismatch, mismatch at minor histocompatibility loci is expected to contribute to inter-patient differences in GVHD development. This may explain why a genetic predisposition towards high IL10-production (GG) was most protective in MUD graft recipients, as greater alloreactivity due to minor histocompatibility mismatch is expected than in MRD transplants [29,30]. In a meta-analysis on the association between IL-10 variants and GVHD, the authors found no significant association between donor IL-10 -1082G/A and grade II–IV aGVHD [40]. This may potentially be explained by the fact that 4/5 included studies consisted of only MRD transplants. The fifth study in the analysis, which was the only study conducted in MUD donors, did in fact find a significant association between -1082G/A and aGVHD [14]. It has also been previously reported that the incidence of aGVHD is lower in Asian populations than in Caucasian populations for both MUD and MRD transplants [41,42]. Although it was initially postulated that this may be a result of reduced diversity at HLA and minor histocompatibility loci, an alternate explanation may be related to the substantially higher frequency of high-IL10-producing alleles in Asian populations [13,43,44].

The current study is also the first, to date, to have conducted assays to verify the degree of donor chimerism post-transplant. Our findings clearly demonstrate that protection against aGVHD is attributed to the donor’s IL10 genotype, as evidenced by three main factors: (i) all patients in this study received myeloablative conditioning, (ii) recipient IL-10 SNPs were not found to be associated with differences in GVHD outcomes, and (iii) chimerism studies demonstrated high donor chimerism in the T- and myeloid cells at 1 month post-transplant. However, in contrast to our findings, most clinical studies have reported that the recipient [12,13,16,20,26,27], rather than donor [14,21,28], IL-10 genotype is associated with GVHD development. Others still have found no association between either donor or recipient IL-10 genotype and GVHD [15,22,23,24,25]. Several possible explanations exist for this discordance. All studies that reported results different to our own either contained some patients that received reduced-intensity (RIC) [16,22,23,24,25], rather than myeloablative, conditioning (MAC) or did not report sufficient information in their conditioning protocol to differentiate between the use of RIC vs. MAC [11,15,21,26]. In the RIC setting, the recipient’s genotype could play a more prominent role in the pathogenesis of GVHD, thereby explaining why recipient rather than donor IL-10 genotype may impact GVHD in this setting. 

IL-10 is also known to play a role in the maintenance of CD4^+^CD25^+^ T-regulatory cells (Tregs) and the induction of type-I Tregs, both of which can suppress GVHD [45,46,47,48]. In murine models, the depletion of CD4^+^CD25^+^ Tregs from the donor graft or in the recipient following HCT has been shown to promote GVHD development [45,49,50,51]. It has been shown that the infusion of donor, but not recipient, CD4^+^CD25^+^ Tregs can rescue mice from lethal GVHD in an IL-10-dependent manner [47], and that donor-graft-derived, but not recipient-derived, IL-10 is critical for Treg-mediated suppression of GVHD [52]. These findings further support the importance of the donor IL-10 genotype in GVHD pathogenesis. 

This study was subject to a few limitations. First, clinical outcomes were studied prospectively only in the validation cohort, and the discovery cohort findings are subject to all of the limitations of a retrospective analysis. However, the major findings of the study were consistent in both cohorts. Second, patients in this study were fairly homogeneous with respect to transplant conditioning, GVHD prophylaxis, and supportive care practices, which aimed to minimize confounding factors in the analysis. Further investigations are warranted to determine the generalizability of these findings to other transplant settings. Third, this study included only 10/10 HLA-matched transplants, again, with the aim of minimizing confounding factors. Since the conception of the study, evidence has emerged supporting the role of other HLA-loci including HLA-DPB1 in GVHD pathogenesis [53,54]. Our group has since shown that mismatch at HLA-DPB1 does not appear to impact GVHD incidence in 10/10 HLA-matched patients treated similarly to those in this study [55]; however, it is nonetheless a limitation that these loci were not considered in our analysis. Additionally, as infection-related non-relapse mortality is an uncommon complication, experienced by only 15/315 (5%) of patients in this study, sample size within IL-10-1082 genotype subgroups was not amenable to a formal statistical comparison. Finally, this study did not assess pre-transplant IL-10 mRNA expression and serum protein levels in recipients and is unable to comment on whether low-IL10-producing recipients become high-IL10-producers (or vice-versa) post-HCT. 

In summary, a genetic predisposition to high IL10 production (IL-10-1082GG) in HCT donors appears to confer protection against aGVHD in recipients. The protective effect is predominant in MUD grafts and may occur through IL10-mediated suppression of alloreactive CD8^+^ T-cells responses, which are known to play a key role in aGVHD pathogenesis. Although the role of IL-10 polymorphisms has been reported both in the context of HCT and solid organ transplantation [33,56], this is the first study, to date, to present corroborating evidence at the genome to cellular level for a non-HLA immunogenetic variant associated with GVHD outcomes. Our study demonstrates that pre-transplant assessment of IL-10 genotypic variants may allow for improved donor selection and the identification of high GVHD-risk patients who are candidates for intensified prophylaxis. A prospective multi-center study is needed to validate these findings and more conclusively determine which patients may benefit from IL-10 immunogenetic matching. As the production of IL-10 is tissue-specific (paracrine), future directions may also include investigations of the role of IL-10 SNPs in organ-specific GVHD. 

## 4. Materials and Methods

### 4.1. Patients and Transplantation

A total of 630 adult subjects were recruited for this study, consisting of 315 donors and 315 recipients who first underwent myeloablative HCT. Donor–recipient pairs were randomly allocated into one of two cohorts for biomarker discovery (*n* = 171) and validation (*n* = 144). All donor–recipient pairs included in this study received an 10/10 HLA-matched graft from a matched related donor (MRD; *n* = 158) or a matched unrelated donor (MUD; *n* = 157). This restriction aimed to reduce HLA mismatch as a potential confounding variable in the analysis. Inclusion and exclusion criteria are presented in the study flow diagram (Appendix A). The clinical and demographic characteristics of donor–recipient pairs are summarized in Table 1. Importantly, the discovery and validation cohorts were balanced in measured characteristics.

All study patients received Fludarabine (250 mg/m^2^; administered 50 mg/m^2^/day across days −6 to −2), pharmacokinetically adjusted intravenous Busulfan (12.8 mg/kg; administered 3.2 mg/kg/day across days −5 to −2), and anti-thymocyte globulin (ATG, 4.5 mg/kg; administered across days −2 (0.5 mg/kg), −1, and 0 (2 mg/kg)) as part of the transplant conditioning regimen. Sixty-nine percent (*n* = 217) of recipients were also administered with total body irradiation (TBI; 400 cGy in two fractions on days −1 or 0 preceding graft infusion). Additional GVHD prophylaxis was with methotrexate on days 1, 3, 6, and 11, and cyclosporine A from day −1 to 84, with taper between days 56 and 84 unless the patient had developed GVHD [57]. Supportive care included cytomegalovirus (CMV)-safe blood products and pre-emptive therapy with valganciclovir [58], zoster prophylaxis with valacyclovir until 2 years (longer if immunosuppressive therapy was used) [32], Pneumocystis/pneumococcal prophylaxis with cotrimoxazole until 6–12 months (longer if immunosuppressive therapy was used) [59], and Candida prophylaxis with fluconazole until 1 month post-transplant [57]. Clinically significant GVHD (csGVHD, defined as grade 2–4 acute GVHD [60] or moderate–severe chronic GVHD [61]) was treated with a corticosteroid, with or without an additional immunosuppressive modality. Supportive care practices were uniform across patients, as previously described in Ousia et al. (2020) [57].

### 4.2. DNA Extraction and Genotyping

High-resolution HLA typing was performed as described in Khan et al. (2012) [58]. Genotyping of cytokine-related genes was performed for the subjects of the discovery cohort (*n* = 171) by sequence-specific primer polymerase-chain reaction (SSP-PCR) using a commercially available Cytokine Genotyping panel (Invitrogen™ Cytokine Genotyping Kit). A total of 22 SNPs located in the promoter, exonic, or untranslated regions of 13 different cytokine or cytokine-receptor genes were analyzed in the discovery cohort and SNPs that were found to be significantly associated with outcomes were then analyzed in the validation cohort (Appendix A). Donors of the validation cohort (*n* = 144) and HCT recipients (*n* = 315) were genotyped for the SNPs located in the promoter region of IL-10 (-1082G/A, -819C/T, and -519C/A) by resequencing of the PCR product. As studies have previously described an association between recipient IL-10 genotype and GVHD [12,13,16,20,26,27], we also assessed this association through the sequencing of recipient samples. A detailed description of the assay is available in the Appendix A.

### 4.3. Determination of IL-10 Serum Levels

Serum samples from a stored serum repository were collected at 1 week (7 days), 1 month (28 days), 2 months (56 days), and 3 months (84 days) post-transplant from 217 randomly selected study patients. At the time of collection, the serum was separated from peripheral blood samples and centrifuged within 2 h after blood collection before being stored within a −80 °C freezer in our sample repository. IL-10 levels were determined by means of a quantitative sandwich enzyme-linked immunosorbent assay (ELISA) technique using a Human IL-10 Platinum ELISA Kit (eBioscience, Vienna, Austria).

### 4.4. Determination of IL-10 mRNA Expression

IL-10 gene-expression analysis was performed in 140 randomly selected study patients using the nCounter^®^ Human Immunology Panel (Nanostring Technologies, Seattle, WA, USA). Total RNA was extracted from 5 million peripheral blood mononuclear cells at day 28. This timepoint was chosen as the median date of clinical diagnosis for aGVHD at our center is 32 days post-transplant [57]. RNA integrity was assessed using a 2100 Bioanalyzer and RNA nanochip assay (Agilent Technologies, Wilmington, DE, USA). A total of 100 ng total RNA was used as the input for the Nanostring assay, which was conducted according to manufacturer’s protocol. Analysis and normalization of raw transcript counts was conducted in nSolver Analysis software V2.0 (Nanostring Technologies, Seattle, WA, USA). For a detailed description of the assay, refer to the Appendix A.

### 4.5. Donor Chimerism Assessment

DNA was extracted from peripheral blood cells from the donor and recipient pre-transplant using the QIAamp DNA Micro-Kit (QIAGEN Inc., Hilden, Germany) for the initial determination of donor and recipient genotype. Peripheral blood samples were obtained from recipients at one month post-transplant and sorted for CD3^+^ T-cells and CD13^+^CD33^+^ myeloid cells using the BD FACS Aria cell sorter. Chimerism assessment was conducted using the Identifiler STR (short tandem repeat) panel kit (AmpFISTR^®^ Identifiler^®^ Plus PCR Amplification Kit) from Applied Biosystems, ThermoFisher Scientific (Woolston, Warrington, UK) (Appendix A). 

### 4.6. Enumeration of Immune Cell Subsets by Multicolor Flow-Cytometry

Our group has previously shown that in the setting of ATG-conditioned myeloablative HCT, aGVHD is preceded by high counts of CD4^+^ and CD8^+^ T-cells [32]. Therefore, we sought to assess the impact of GVHD-associated cytokine gene variants and the recovery of T-cells post-HCT. Data was extracted from two previously published studies from our group that measured blood-cell subsets in patients at the study center [32,59]. A total of 122 alloHCT recipients from those studies were also part of the current study, and in these patients, we correlated data on immune-cell subsets from these patients with IL-10 gene variants. Multi-color flow cytometry-based enumeration of T-cell subsets at 1 week, 1 month, 2 months, and 3 months post-transplant were included in the analysis. Details of the assay are available in the Appendix A Online.

### 4.7. Outcomes and Statistical Analysis

GVHD was diagnosed according to historical criteria (acute; aGVHD- if onset by day 100, and chronic; cGVHD- if present after day 100). If the patient was diagnosed with distinctive features of cGVHD before day 100, then it was classified as Cgvhd [32]. aGVHD was classified from grade I–IV as described by Glucksberg et al [60] and grading was according to the 1994 consensus conference [61]. cGVHD was graded as per Seattle [62] and NIH [63] criteria. Relapse was defined using standard criteria (e.g., >5% blasts in the case of acute leukemia). Clinically significant GVHD (sGVHD) was defined as grade II–IV aGVHD or moderate–severe cGVHD NST.

Statistical analysis of pre-defined cause-specific outcomes (aGVHD, cGVHD, sGVHD, and relapse) was conducted using competing-risk regression (Fine-Gray regression). Competing risks are as described in Appendix A. Kaplan–Meier estimation and Cox regression were used to evaluate the association of genetic variants with overall survival (OS). Variables of interest for the multivariate regression modeling were selected based on associations previously described in the literature. We first computed the statistical significance of associations between covariates and HCT outcomes using univariate analysis (Appendix A). Covariates that were statistically significant (*p* < 0.05) were included in multivariate regression analyses. Time was defined as days from transplantation to the relevant failure event, censured to last medical contact. The association of IL-10 serum cytokine levels, mRNA expression, and immune reconstitution of T-cell subsets with IL-10 donor polymorphisms were compared using a non-parametric test for independent sampling. All analyses were performed using STATA IC 11.2 software. A two-tailed *p*-value of <0.05 was considered statistically significant.

## Figures and Tables

**Figure 1 ijms-23-15888-f001:**
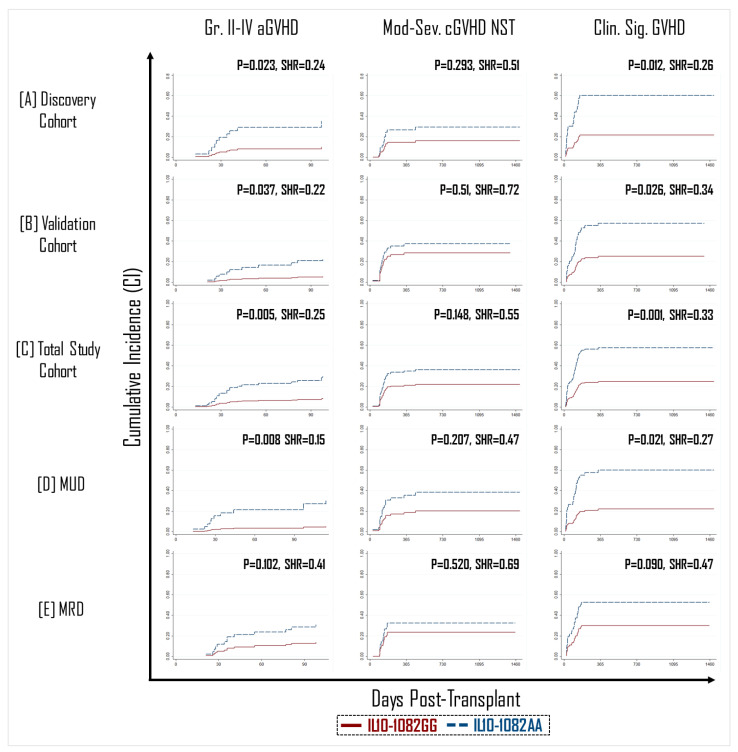
Recipients of IL10-1082GG positive graft had a three-fold lower incidence of grade 2–4 acute GVHD (aGVHD) (**left panel**) and clinically significant GVHD (**right panel**) than recipients of an IL10-1082AA positive graft across both the randomly allocated discovery (**A**) and validation (**B**) cohorts, as well as in the total study cohort (**C**). The protective effect of the IL10-1082GG variant was more pronounced in matched unrelated donor (MUD) HCT (**D**), than matched related donor (MRD) HCT (**E**). No impact of IL10-1082 polymorphisms was observed on the incidence of moderate–severe chronic GVHD (cGVHD) (**middle panel**).

**Figure 2 ijms-23-15888-f002:**
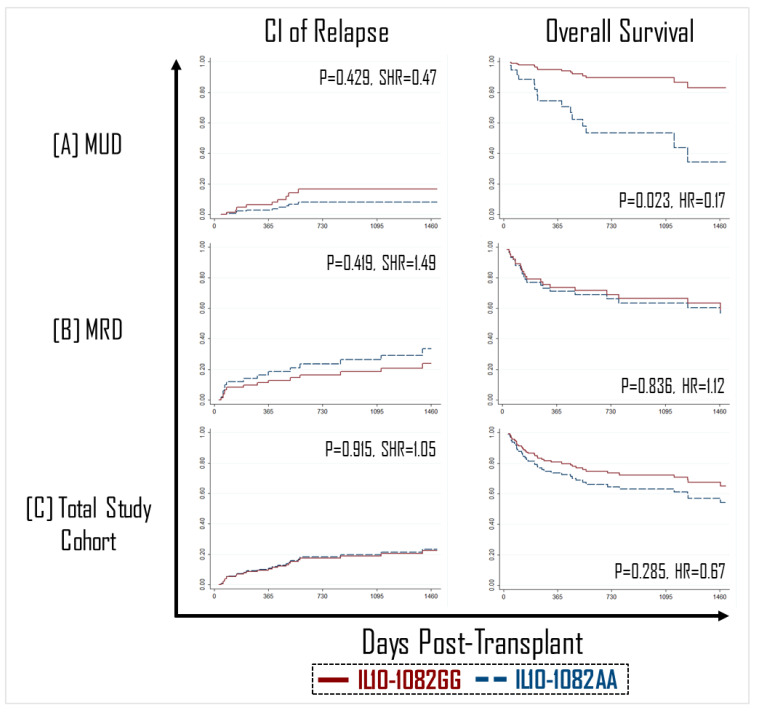
No impact of IL10-1082 polymorphisms was observed on the cumulative incidence (CI) of relapse (**left panel**, (**A**–**C**)). HCT recipients with an IL10-1082GG graft had improved overall survival in matched unrelated donor (MUD) transplants (**right panel**, (**A**)), but not matched unrelated donor (MRD) (**right panel**, (**B**)).

**Figure 3 ijms-23-15888-f003:**
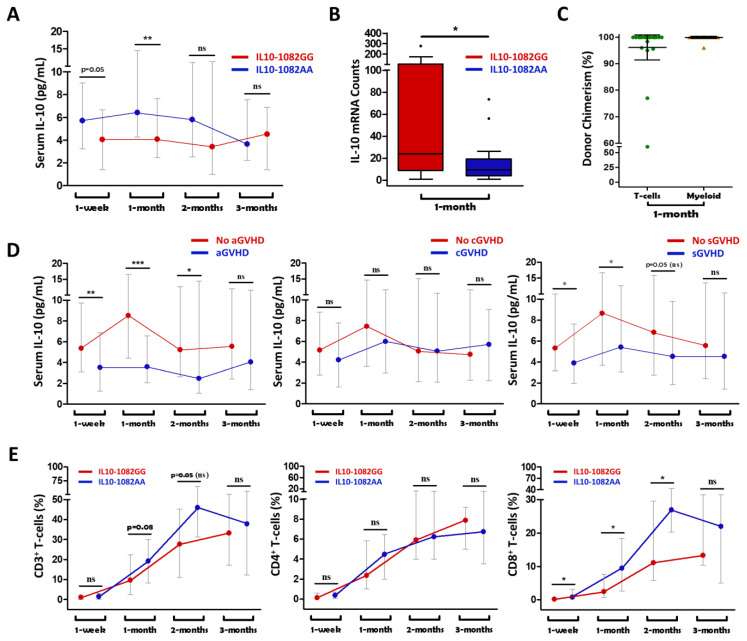
Recipients of an IL10-1082GG graft had greater serum IL10 levels at 1 week and 1 month post-transplant compared to recipients of an IL10-1082A graft (**A**). Recipients of IL10-1082GG grafts demonstrated higher IL10 mRNA counts at 1 month post-transplant (**B**). Complete (>95% donor) chimerism in both T-cells and myeloid cells supports a predominant donor-IL10-genotype effect on IL10 mRNA and serum cytokine levels (**C**). Higher serum IL10 levels were associated with a lower incidence of aGVHD and sGVHD, but not cGVHD (**D**). Recipients of an IL10-1082GG graft had a relatively attenuated population of CD8+ T-cells compared to IL10-1082AA graft recipients (**E**). “*” = *p* < 0.05; “**” = *p* < 0.01; “***” = *p* < 0.001; “ns” = not statistically significant.

**Table 1 ijms-23-15888-t001:** Demographic and clinical characteristics of donor–recipient pairs.

Characteristic	Total Patients	Discovery Cohort	Validation Cohort	*p*-Value *
No. Donor–Recipient Pairs	315	171	144	-
Median Patient Age, years (range)	50 (18–66)	50 (18–66)	49 (18–66)	1.00
Median Donor Age, years (range)	37 (12–69)	37 (12–67)	36 (12–69)	0.59
Donor/Recipient Sex				0.66
Male/Male	120 (38%)	64 (37%)	56 (39%)	
Male/Female	82 (26%)	41 (24%)	41 (28%)	
Female/Male	60 (19%)	34 (20%)	26 (18%)	
Female/Female	53 (17%)	32 (19%)	21 (15%)	
Graft Source				0.12
PBSC	293 (93%)	163 (95%)	130 (90%)	
BM	22 (7%)	8 (5%)	14 (10%)	
Donor Type				0.43
8/8 HLA MRD	158 (50%)	82 (48%)	76 (53%)	
8/8 HLA MUD	157 (50%)	89 (52%)	68 (47%)	
Conditioning				0.70
Flu-Bu-TBI	217 (69%)	120 (70%)	97 (67%)	
Flu-Bu	98 (31%)	51 (30%)	37 (33%)	
GVHD Prophylaxis				-
MTX + CNI + ATG	315 (100%)	171 (100%)	144 (100%)	
Disease Type				0.06
AML	109 (34%)	54 (34%)	45 (31%)	
ALL	46 (14%)	25 (14%)	21 (15%)	
MDS	30 (9%)	12 (9%)	18 (13%)	
CLL	16 (5%)	12 (5%)	4 (3%)	
CML	23 (7%)	9 (7%)	14 (10%)	
Lymphoma	32 (10%)	14 (8%)	18 (13%)	
Other	69 (21%)	45 (26%)	24 (17%)	
Disease Risk **				0.82
Low Risk	152 (48%)	84 (49%)	68 (47%)	
High Risk	163 (62%)	87 (52%)	76 (53%)	
CMV Serostatus				0.20
D−R−	96 (30%)	51 (30%)	45 (30%)	
D+R−	34 (11%)	24 (14%)	10 (7%)	
D−R+	76 (24%)	39 (23%)	37 (25%)	
D+R+	113 (35%)	55 (32%)	58 (38%)	
Unknown	4 (1%)	3 (2%)	1 (1%)	
EBV Serostatus				0.45
D−R−	4 (1%)	1 (1%)	3 (2%)	
D+R−	17 (5%)	7 (4%)	10 (7%)	
D−R+	24 (8%)	15 (9%)	9 (6%)	
D+R+	257 (82%)	142 (83%)	115 (80%)	
Unknown	13 (4%)	6 (4%)	7 (5%)	
Median Followup, days (range)	703 (6–2632)	689 (6–2554)	718 (14–2632)	0.92

* Statistical difference calculated using Mann–Whitney–Wilcoxon test, Fisher’s exact test, or chi-squared test. ** Good (or low) risk disease was defined as acute leukemia in first remission, chronic myelogenous leukemia in first chronic or accelerated phase, or myelodyplastic syndrome if <5% marrow blasts of aplastic anemia. All other diseases/disease stages were considered high risk. Abbreviations: PBSC = peripheral blood stem cells; BM = bone marrow; HLA = human leukocyte antigen; MRD = matched related donor; MUD = matched unrelated donor; Flu = fludarabine; Bu = busulfan; TBI = total body irradiation; MTX = methotrexate; CNI = calcineurin inhibitor; ATG = anti-thymocyte globulin; AML = acute myeloid leukemia; ALL = acute lymphoid leukemia; MDS = myelodysplastic syndrome; CLL = chronic lymphocytic leukemia; CML = chronic myeloid leukemia; MF = myelofibrosis; CMV = cytomegalovirus; D = donor; R = recipient; EBV = Epstein–Barr virus.

## Data Availability

Data analysis was performed by G.T. and R.A.K. and all authors had access to the primary data. Requests for data sharing will be accommodated where possible in accordance with institutional policy and regional legislation.

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
