# Peer review of "Donor Genetic Predisposition to High Interleukin-10 Production Appears Protective against Acute Graft-Versus-Host Disease"

_ijms, 2022, doi:10.3390/ijms232415888_

Round 1

Reviewer 1 Report

The paper describes an interesting association between a polymorphic variant in IL10 gene promoter od HSCs donors and post-HSCT outcomes in a group of HSC recipients. Overall, I find this work very interesting.

In the materials and methods section, I would recommend to inform the readers of more clinical characteristics of the patients considered in the analysis, namely: GVHD prophylaxis after transplant, route of Busulfan administration (iv?) in line 313, diagnosis - which impacts on the relapse risk, age, HCT-CI or other comorbidity index, DRI or other disease risk score. I read reference to "Table 1" in the text, but I do not see this table in the file nor in the supplementary material. Is the ATG dose identical in patients receiving graft from MUD and MSD as reported in line 314?

In addition, since the paper analyses the impact of IL10 "secretion profile" on the clinical transplant outcomes, I would like to know if any difference in infectious complications was observed in the patients considered, along with the other outcomes analyzed.

In the Results section, follow-up time of patient cohorts is missing (are the OS and GVHD analysis censored to last follow-up?). I think this information is relevant to describe survival and GVHD outcomes.

Reviewer 2 Report

The article is dedicated to genetic predisposal for graft-versus-host disease (GVHD) after allogeneic hematopoietic stem cell transplantation. The studies of non-HLA genes as GVHD factors are performed since last 20 years. In this paper, however, the authors have shown a protective effect of -1082GG IL10 polymorphism against moderate/severe (grade II-IV) acute GVHD compared to IL10-1082AA recipients, along with other effects of this SNP. The study was performed in relatively large cohort with 315 donor-recipient pairs, 10/10 HLA matched, uniform conditioning regimens). The authors evaluated three well-known SNPs in the IL-10 gene promoter, with clear effect found for SNP at IL10 -1082. This association proved to be more pronounced in matched unrelated transplants. Generally, the effect of the donor genotype was more expressed. The -1082GG genotype was shown to be associated with higher IL10-production and lower CD8+ T cell response, thus explaining the genetic/clinical associations observed.

Some remarks still exist:

 Results (lane 163-164): one may be cautious about full chimerism of T cell compartment 1 mo post-transplant. In Fig.3C, the scatter of results is rather high, with chimerism <90%  in some cases. In this respect, one should consider the conditioning regimens with fludarabine/busulfan as not fully myeloablative, thus providing partial chimerism in some cases. However, this objection is here not principal.

Materials and methods (lane 349): one should, at least, in brief, describe the method of IL-10 mRNA counting (e.g., what gene served as a reference value, and what formule was used for quantitation of IL-10 gene expression).

 The text should undergo minor copy editing, due to some slips of language.

In general, the manuscript contains original and comprehensive data on clinical and hematological effects of IL-10 promoter variant thus deserving publication with only minor corrections.
